# EZ-STANCE: A Large Dataset for Zero-Shot Stance Detection

## Abstract

Zero-shot stance detection (ZSSD) aims to determine whether the author of a text is in favor of, against, or neutral toward a target that is unseen during training. In this paper, we present EZ-STANCE, a large English ZSSD dataset with 30,606 annotated text-target pairs. In contrast to VAST, the only other existing ZSSD dataset, EZ-STANCE includes both noun-phrase targets and claim targets, covering a wide range of domains. In addition, we introduce two challenging subtasks for ZSSD: target-based ZSSD and domain-based ZSSD. We provide an in-depth description and analysis of our dataset. We evaluate EZ-STANCE using state-of-the-art deep learning models. Furthermore, we propose to transform ZSSD into the NLI task by applying two simple yet effective prompts to noun-phrase targets. Our experimental results show that EZ-STANCE is a challenging new benchmark, which provides significant research opportunities on ZSSD. We will make our dataset and code available on GitHub.

## 1 Introduction

The goal of stance detection is to automatically detect whether the author of a text is in favor of, against, or neutral toward *a specific target* (Mohammad et al., 2016b; Küçük and Can, 2020; AL-Dayel and Magdy, 2021), e.g., public education, mask mandate, or nuclear energy. The detected stance can reveal valuable insights relevant to significant events such as public policy-making and presidential elections.

Earlier research has concentrated on two types of stance detection tasks: in-target stance detection, in which models are trained and evaluated using data from the same set of targets, e.g., both train and test contain data about "Donald Trump" (Hasan and Ng, 2014; Mohammad et al., 2016b; Graells-Garrido et al., 2020), and cross-target stance detection, where the models are trained on source targets that are related to, but distinct from, the destination targets (Augenstein et al., 2016; Wei and Mao, 2019), which remain unseen during training (e.g., destination target is "Donald Trump" whereas source target is "Hillary Clinton"). However, it is unrealistic to incorporate every potential or related target in the training set. As such, zero-shot stance detection (ZSSD) has emerged as a promising direction (Allaway and McKeown, 2020) to evaluate classifiers on a large number of unseen (and unrelated) targets. ZSSD is more related to real-world scenarios and has consequently started to receive significant interest recently (Liu et al., 2021; Luo et al., 2022; Liang et al., 2022b).

Despite the growing interest in ZSSD, the task still exhibits several limitations. First, the VAST dataset (Allaway and McKeown, 2020) which is the only existing ZSSD dataset, contains only noun phrase targets. Yet, in real-world scenarios, stances are often taken toward both noun phrases (Mohammad et al., 2016b; Glandt et al., 2021) and claims (Ferreira and Vlachos, 2016; Derczynski et al., 2017). We observe that models trained exclusively on noun-phrase targets do not perform well on claim targets (or vice versa), due to the mismatch between the training and test data. The need to incorporate both types of targets for ZSSD has been relatively overlooked. Second, VAST is designed solely to detect the stance of unseen targets, but these unseen targets at the inference stage originate from the same domain as the training targets (in-domain), possessing similar semantics, which makes the task less challenging. Third, despite being instrumental for the development of zero-shot stance detection, VAST generates data for the neutral class by randomly permuting existing documents and targets, leading to a lack of semantic correlation between the two. Models can easily detect these patterns, consequently diminishing the complexity of the task.

In an effort to address the aforementioned limitations and spur research in ZSSD, we present EZ-

| Tweet | Nuclear Energy is a much safer and cost-efficient source of energy than coal and oil and people should be using it! |
|---|---|
| Stance/Noun-phrase targets | **Favor** / Nuclear Energy
**Against** / Coal |
| Stance/Claim targets | **Favor** / Compared with traditional energy such as coal and gasoline, nuclear brings more security and is more economical.
**Against** / Don't play with nuclear! We should stick with coal and fossil fuels.
**Neutral** / Nuclear Energy will soon be the only energy left in the market. Coal and oil are outdated. |

Table 1: Examples of noun-phrase targets and claim targets for a tweet in the "**Environmental Protection**" domain of our EZ-STANCE dataset.

STANCE, a large **E**nglish **Z**ero-shot stance detection dataset collected from Twitter. In contrast with VAST, EZ-STANCE is, to our knowledge, the first large ZSSD dataset that captures both noun-phrase targets and claim targets, covering a more diverse set of targets. Moreover, EZ-STANCE includes two real-world scenarios for zero-shot stance detection, namely target-based and domain-based ZSSD. **Subtask A: target-based zero-shot stance detection**. This subtask is the same as the traditional ZSSD task (Allaway and McKeown, 2020), where stance detection classifiers are evaluated using a large number of completely unseen (and unrelated) targets, but from the same domains (in-domain). **Subtask B: domain-based zero-shot stance detection**. Subtask B is our proposed ZSSD task where stance detection classifiers are evaluated using a large number of unseen targets from completely new domains (out-of-domain). Furthermore, in EZ-STANCE, annotators manually extract targets from each tweet to form the neutral class, ensuring semantic relevance to the tweet content. An example tweet from our dataset along with corresponding noun-phrase and claim targets and their stance are shown in Table 1. As we can see from the table, the author of the tweet is in favor of the noun-phrase target "Nuclear Energy" and against "Coal". The author also opposes claim target 2, whose main idea is to refute the need for nuclear energy.

In summary, our contributions are as follows: 1) We present EZ-STANCE, a unique large zero-shot stance detection dataset, composed of 30,606 annotated English tweet-target pairs. EZ-STANCE is 1.9 times larger than VAST (Allaway and McKeown, 2020), which is the only large existing ZSSD dataset for English. We provide a detailed description and analysis of our dataset; 2) We consider a more diverse set of targets including both noun phrases and claims in EZ-STANCE (see Table 1); 3) We include two challenging ZSSD subtasks in EZ-STANCE: target-based zero-shot stance detection and domain-based zero-shot stance detection; 4) We establish baseline results using both traditional models and pre-trained language models; 5) We propose to formulate stance detection into the task of natural language inference (NLI) by applying two simple yet effective prompts for noun-phrase targets. Our results and analysis show that EZ-STANCE is a challenging new benchmark.

## 2 Related Work

Most earlier research is centered around in-target stance detection where a classifier is trained and evaluated on the same target (Zarrella and Marsh, 2016; Wei et al., 2016; Vijayaraghavan et al., 2016; Mohammad et al., 2016b; Du et al., 2017; Sun et al., 2018; Wei et al., 2018; Li and Caragea, 2019, 2021). However, the challenge often arises in gathering enough annotated data for each specific target, and traditional models perform poorly when generalized to unseen target data. This spurred interest in investigating cross-target stance detection (Augenstein et al., 2016; Xu et al., 2018; Wei and Mao, 2019; Zhang et al., 2020), where a classifier is adapted from different but related targets. However, cross-target stance detection still requires prior human knowledge of the destination target and how it is related to the training targets. Consequently, models developed for cross-target stance detection are still limited in their capability to generalize to a wide range of unseen targets (Liang et al., 2022b). Zero-shot stance detection (ZSSD) which aims to detect the stance on a large number of unseen (and unrelated) targets has received significant interest in recent years. Allaway and McKeown (2020) developed VAried Stance Topics (VAST), the only existing dataset for ZSSD that encompasses thousands of noun-phrase targets. Some ZSSD models have been developed based on VAST (Liu et al., 2021; Liang et al., 2022a,b; Luo et al., 2022). In contrast with VAST, we include two types of ZSSD subtasks in EZ-STANCE. Target-based ZSSD is the same as the VAST setting. For domain-based ZSSD, classifiers are evaluated on unseen targets from completely new domains, which is a more challenging task. Moreover, data for the neutral class in the VAST dataset is generated by randomly permuting existing documents and targets, resulting in easy-to-detect patterns. Comparatively, in

| Domain | | Query Keywords |
|---|---|---|
| Covid Epidemic | CE | epidemic prevention, living with covid, herd-immunity, WFH, booster, vaccine, mask mandate, FDA, post-covid, Fauci |
| World Events | WE | world news, Ukraine, Russia, migrant, NATO, China, Mideast, negative population growth, terrorism |
| Education and Culture | EdC | public education, pop culture, cultural output, home schooling, AI assistance writing, arming teachers, private education, international student |
| Entertainment and Consumption | EnC | prices, gasoline price, online shopping, TikTok, iPhone, Reels, Disney, medical insurance, ethical consumption, vegetarian |
| Sports | S | World Cup, NBA, men's football, women's football, NCAA, MLB, NFL, WWE |
| Rights | R | gender equality, equal rights, women's rights, LGBTQ, BLM, doctors and patients, racism, Asian hate, gun control |
| Environmental Protection | EP | climate change, clean energy, environmental awareness, environmental protection agency, shut down coal plants, nuclear energy, electric vehicle |
| Politics | P | government, republican, reform, leftists, democrat, democracy, right-wing, politic, presidential debate, presidential election, midterm election |

Table 2: The domains used in our dataset and the selected query keywords for each domain.

our dataset, targets of the neutral data are extracted based on the documents, ensuring strong semantic relevance to the document content. We compare our EZ-STANCE dataset with previous stance detection datasets in Appendix A.

Target-specific stance detection is the prevalent stance detection task (ALDayel and Magdy, 2021), whose goal is to determine the stance for a target, which could be a figure or controversial topic (Hasan and Ng, 2014; Mohammad et al., 2016a; Zotova et al., 2020; Conforti et al., 2020a,b). In contrast, claim-based stance detection aims to predict the stance toward a specific claim, which could be an article's headline or a reply to a rumorous post (Qazvinian et al., 2011; Derczynski et al., 2015; Ferreira and Vlachos, 2016; Bar-Haim et al., 2017; Derczynski et al., 2017; Gorrell et al., 2019). Nonetheless, insufficient attention has been paid to integrating both noun-phrase targets and claim targets into a single dataset. In contrast, our dataset accommodates data with both noun-phrase targets and claim targets (see examples in Appendix B).

## 3 Dataset Construction

In this section, we detail the creation of EZ-STANCE, our large English ZSSD dataset consisting of 30,606 annotated instances covering a comprehensive range of domains.

### 3.1 Data Collection

Our data are collected using the Twitter API, spanning from May 30th, 2021 to January 20th, 2023. In alignment with previous works (Mohammad et al., 2016b; Glandt et al., 2021; Li et al., 2021), we crawl tweets using query keywords. To cover a wide range of domains on Twitter, we begin with the domain names from the *Explore* page of Twit-

ter as keywords for data crawling (e.g., Covid epidemic, education, etc.). After we collect our initial set, we gradually expand the keywords set for the next round by including the most frequent words as supplementary keywords. The full list of keywords that we used for crawling is provided in Appendix C. In total, we collect 50,000 tweets.

After this, we perform keyword filtering to eliminate keywords that are not suitable for stance detection. Our keyword filtering is performed in the following steps: 1) We manually detect a subset of tweets crawled using each keyword and we remove keywords that are frequently associated with promotional content (e.g., YouTuber, live shopping, etc.), whose main purpose is for product/people promotion instead of addressing controversial topics; 2) Keywords that people predominantly hold single stances on are filtered out, e.g., pollution, crime, delicious food, etc. This is because models would simply learn the correlation between the keywords and the stance and predict stances based solely on keywords instead of the content of tweets and targets. After filtering, we select 72 keywords covering controversial topics. We summarize the 72 keywords into 8 domains: "Covid Epidemic" (CE), "World Events" (WE), "Education and Culture" (EdC), "Entertainment and Consumption" (EnC), "Sports" (S), "Rights" (R), "Environmental Protection" (EP), and "Politics" (P). Table 2 shows the domains and query keywords in each domain.

### 3.2 Preprocessing

To ensure the quality of our dataset, we perform the following preprocessing steps: 1) We remove tweets with less than 20 or more than 150 words. According to our observations, tweets with less than 20 words are either too easy or cannot include enough information to express stances toward mul-

| Domain | Noun-phrase targets | | | Claim targets | | |
|---|---|---|---|---|---|---|
| | Con | Pro | Neu | Con | Pro | Neu |
| CE | 625 | 505 | 488 | 862 | 862 | 862 |
| WE | 557 | 367 | 540 | 772 | 772 | 772 |
| EdC | 395 | 538 | 436 | 731 | 731 | 731 |
| EnC | 429 | 601 | 703 | 945 | 945 | 945 |
| S | 125 | 516 | 500 | 625 | 625 | 625 |
| R | 574 | 660 | 340 | 786 | 786 | 786 |
| EP | 318 | 624 | 350 | 611 | 611 | 611 |
| P | 758 | 538 | 507 | 872 | 872 | 872 |
| Overall | 3,781 | 4,349 | 3,864 | 6,204 | 6,204 | 6,204 |

Table 3: Label distribution for noun-phrase targets and claim targets in each domain from our dataset. Con, Pro, Neu represent against, favor, and neutral, respectively.

tiple targets. Tweets with more than 150 words usually contain links to external content; 2) We remove duplicates and retweets; 3) We keep only tweets in English; 4) We filter out tweets containing advertising contents (e.g., scan the QR code, reply or DM me, sign up, etc.); and 5) We remove emojis and URLs as they may introduce noise. We randomly select around 86 tweets for each keyword, obtaining 6204 tweets for annotation.

### 3.3 Data Annotation

The target and stance annotations of our dataset are gathered through Cogitotech,[1] a data annotation company that provides annotation services for big AI companies (e.g., OpenAI, AWS, etc.). To ensure high-quality annotations, we apply rigorous criteria: 1) Annotators should have a minimum education qualification of college graduation; 2) The annotators' native language must be English. Moreover, we randomly sample 10% of each annotator's annotations to perform quality checks and discard annotations from an annotator if the acceptance rate is lower than 90%. This data is re-sent to other qualified annotators for labeling. The stance label distribution for both noun-phrase and claim targets for each domain is shown in Table 3.

#### 3.3.1 Annotation for Noun-Phrase Targets

The annotation for noun-phrase targets is performed in the following two steps. In step 1, one annotator is asked to identify a minimum of 2 targets from each given tweet. Annotators are given the following instructions: *From each tweet, please identify at least 2 noun-phrase targets. Targets should meet the following criteria: 1) Targets should be the principal subject of the tweet rather than minor details; 2) Targets should represent widely discussed topics where different stances are exhibited;*

[1]https://www.cogitotech.com/

*3) Targets where people often express the same stance should be avoided, e.g., violence abuse.* In step 2, we instruct 3 annotators to assign a stance label to each tweet-target pair, using the following instructions: *Imagine yourself as the author of the tweet, please annotate the stance that you would take on this given target as "Favor", "Against", or "Neutral".* After the annotations are completed, we determine the stance for each tweet-target pair by using the majority vote amongst the three annotators. For 6,204 tweets, we obtain 11,994 annotated instances (around 2 targets per tweet). The inter-annotator agreement measured using Krippendorff's alpha (Krippendorff, 2011) is 0.63, which is higher than VAST (0.427).

#### 3.3.2 Annotation for Claim Targets

The annotation for claim targets aims to collect three claims, to which the tweet takes favor, against, and neutral stances, respectively. We provide the following instructions: *Based on the message that you learned from the tweet, write the following three claims: 1) The author is definitely in favor of the point or message of the claim (favor); 2) The author is definitely against the point or message from the claim (against); 3) Based solely on the information from the tweet, we cannot know whether the author definitely supports or opposes the point or message of the claim (neutral).* To make this task more challenging, we establish a set of extra requirements: First, claims labeled with *favor* must not replicate the tweet verbatim. Second, claims labeled with *against* should not merely negate the tweet content (e.g., adding "not" before verbs). Models could easily detect such linguistic patterns and predict stances without learning the content of tweet-claim pairs.

For quality assurance, we hide the stance labels for a subset of tweet-claim pairs and ask another group of annotators (who did not write the claims) to annotate the stance. The two groups agree on 95% of the times. This result indicates high-quality generations of the claim targets and stance labels. In total, we obtain 18,612 tweet-claim pairs.

### 3.4 Dataset Split

We partition the annotated data into training, validation, and test sets for both target-based ZSSD (subtask A) and domain-based ZSSD (subtask B). For subtask A, we split the dataset in alignment with the VAST dataset (Allaway and McKeown, 2020): the training, validation, and test sets do not

|  |  | # Examples | | # Unique | | | Avg. Length | | | Lexsim |
|---|---|---|---|---|---|---|---|---|---|---|
|  |  | N | C | N | C | T | N | C | T | (%) |
| Subtask A | Train | 8,705 | 12,264 | 4,842 | 12,248 | 4,088 | 1.8 | 18.4 | 39.8 | - |
|  | Val | 1,667 | 3,081 | 1,578 | 3,078 | 1,027 | 2.3 | 19.2 | 39.1 | 13 |
|  | Test | 1,622 | 3,267 | 1,613 | 3,253 | 1,089 | 2.3 | 18.9 | 39.3 | 12 |
| Subtask B (Covid Epidemic) | Train | 8,498 | 13,167 | 5,875 | 13,151 | 4,389 | 2 | 18.6 | 39.3 | - |
|  | Val | 1,231 | 2,754 | 1,220 | 2,744 | 918 | 2.3 | 18.4 | 39.5 | 11 |
|  | Test | 1,716 | 2,607 | 1,156 | 2,602 | 869 | 1.9 | 18.7 | 41.1 | 10 |

Table 4: Dataset split statistics for subtask A and subtask B ("Covid Epidemic" as the zero-shot domain). N, C, T represent noun-phrase targets, claim targets, and tweets, respectively. Lexsim represents the ratio of LexsimTopics.

share any documents (tweets) and targets with each other. We provide further details of our split approach in Appendix D. The dataset distribution is shown in Table 4. Additionally, we present the average percentage of overlapping tokens in all tweet-target pairs in Appendix E.

For subtask B, we use the data from seven domains (source) for training and validation, and the data from the left-out domain (zero-shot) as the test set. This results in 8 dataset splits for subtask B with one dataset split assigned for each of the eight domains, wherein each domain in turn is used as the test set. To ensure the zero-shot scenario, we remove data with overlapping targets from the source domains in each split. Next, we divide the source domains into training and validation sets, ensuring no overlapping tweets and targets. The statistics when using the "Covid Epidemic" as the zero-shot domain are shown in Table 4. The full statistics of subtask B are shown in Appendix F.

Given the linguistic variations in the noun-phrase target expressions, we investigate the prevalence of *LexSimTopics* (Allaway and McKeown, 2020) between the training and the test set. LexSimTopics is defined as the percentage of targets that possess more than 0.9 cosine similarities with any training targets in the word embedding space (Bojanowski et al., 2017). As shown in Table 4, in Subtask A, we have 12% and 13% *LexSimTopics* in the test set and the validation set, respectively. Whereas for the "Covid Epidemic" domain in subtask B, we only have 10% and 11% *LexSimTopics* for the test and validation sets. This indicates that subtask B poses more challenges as the targets in the training and test sets exhibit more differences. In comparison, the VAST dataset has 16% and 19% *LexSimTopics* in the zero-shot test set and validation set, respectively, which are higher than our dataset.

## 4 Methodology

We now present our approach for converting ZSSD into the natural language inference (NLI) task.

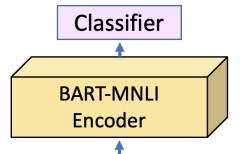

**Premise** *I think electric cars are better for the environment.*
**Hypothesis:** ***The premise entails*** *clean energy!*
**Label:** entailment

**Text:** *I think electric cars are better for the environment.*
**Target:** *clean energy*
**Label:** *favor*

Figure 1: Our approach to transfer ZSSD into NLI.

### 4.1 Problem Definition

Suppose we are given a training set $D^{train}=\{(x_i^{train}, t_i^{train}, y_i^{train})\}_{i=1}^{N_{train}}$ and a test set $D^{test}=\{(x_i^{test}, t_i^{test})\}_{i=1}^{N_{test}}$, where $x_i^{train}$ is a training document (tweet), $t_i^{train}$ is a target in $x_i^{train}$ and $y_i^{train}$ is its label (or stance) $\in$ {*Favor*, *Against*, *Neutral*}. For target-based ZSSD (subtask A), targets in $x_i^{test}$ do not overlap with targets in $x_i^{train}$. For domain-based ZSSD (subtask B), targets in $x_i^{test}$ not only do not overlap with the targets in $x_i^{train}$, but they also belong to a domain that is not seen in $D^{train}$. The objective is to predict the stance given both $x_i^{test}$ and $t_i^{test}$ by training a model on the $D^{train}$.

### 4.2 Transform ZSSD into NLI

We propose to convert the document and target into the premise and hypothesis of NLI, respectively. The task of predicting stance labels (*Favor*, *Against*, or *Neutral*) is transformed into the task of predicting entailment labels (*Entailment*, *Contradiction*, or *Neutral*). In particular, we design two simple yet effective prompt templates to formulate noun-phrase targets into more refined hypotheses, thereby facilitating the model to better leverage the NLI pre-trained model for stance detection. These prompts are: "*The premise entails [target]!*", and "*The premise entails the hypothesis [target]!*" For each noun-phrase target, we randomly apply one of the prompts. Note that we do not apply prompts for claim targets as they already resemble hypotheses quite closely. We fine-tune the BART-large encoder

| | Mixed targets | | | | Noun-phrase targets | | | | Claim targets | | | |
|---|---|---|---|---|---|---|---|---|---|---|---|---|
| | Con | Pro | Neu | All | Con | Pro | Neu | All | Con | Pro | Neu | All |
| BiCE | .539 | .358 | .536 | .478 | .583 | .550 | .453 | .529 | .313 | .346 | .317 | .325 |
| Cross-Net | .504 | .485 | .571 | .520 | .559 | .552 | .466 | .526 | .473 | .448 | .622 | .514 |
| TGA Net | .558 | .564 | .625 | .582 | .641 | .603 | .503 | .583 | .514 | .551 | .687 | .584 |
| BERT | .724 | .732 | .756 | .738 | .669 | .619 | .535 | .608 | .706 | .768 | .872 | .782 |
| RoBERTa | .787 | .785 | .769 | .780 | .712 | .677 | .529 | .639 | .821 | .856 | .881 | .853 |
| XLNet | .767 | .766 | .760 | .764 | .685 | .652 | .531 | .623 | .806 | .841 | .880 | .842 |
| BART-MNLI | .652 | .699 | .632 | .661 | .194 | .531 | .205 | .310 | .789 | .832 | .783 | .801 |
| **BART-MNLI-e** | .816 | .808 | .773 | .799 | .729 | .690 | .542 | .653 | **.858**$^*$ | **.888**$^*$ | **.892**$^*$ | **.879**$^*$ |
| **BART-MNLI-e**$_p$ | **.818**$^*$ | **.813**$^*$ | **.783**$^*$ | **.805**$^*$ | **.739**$^*$ | **.692**$^*$ | **.576**$^*$ | **.669**$^*$ | - | - | - | - |

Table 5: Comparison of different models on EZ-STANCE subtask A. The performance is reported using F1 score for the against (Con), favor (Pro), neutral (Neu), and the $F1_{macro}$ (All). ∗: our approach improves the best traditional baseline at $p < 0.05$ with paired t-test.

(Lewis et al., 2020) pre-trained on MNLI (Williams et al., 2018) to predict the stance. The BART decoder is not included due to memory constraints. As shown in Figure 1. The text "I think electric cars are better for the environment." with stance *Favor* towards the target "clean energy" in ZSSD becomes <premise, hypothesis> as <"I think electric cars are better for the environment.", "The premise entails clean energy!"> with the *Entailment* NLI label.

## 5 Baselines and Models

We evaluate EZ-STANCE using the following baselines. **BiCE** (Augenstein et al., 2016) and **Cross-Net** (Xu et al., 2018) predict the stance using the conditional encoding of BiLSTM. **TGA-Net** (Allaway and McKeown, 2020) captures implicit relations/correlations between targets in a hidden space to assist stance classification. We also consider fine-tuning the base version of state-of-the-art transformer-based models as strong baselines, including **BERT** (Devlin et al., 2019), **RoBERTa** (Liu et al., 2019) and **XLNet** (Yang et al., 2019).

To evaluate NLI pre-trained models for ZSSD, we compare the following methods: **BART-MNLI-e**$_p$: We fine-tune the BART-MNLI encoder using EZ-STANCE dataset with our proposed prompts applied to noun-phrase targets. **BART-MNLI-e**: We fine-tune the BART-MNLI encoder using the original EZ-STANCE dataset without prompts. **BART-MNLI**: We directly use the pre-trained BART-MNLI model with both encoder and decoder without fine-tuning to infer the stance labels for the test set. We show the hyperparameters adopted in our experiments in Appendix G.

## 6 Results

In this section, we first conduct experiments for subtask A (§6.1) and subtask B (§6.2). We then compare our EZ-STANCE with the VAST dataset (§6.3). We also study the impact of different prompts (§6.4). Next, we explore the effects of integrating noun-phrase targets and claim targets into one dataset (§6.5). Lastly, we perform the spuriosity analysis for claim targets (§6.6). Like prior works (Allaway and McKeown, 2020), we employ class-specific F1 scores and the macro-averaged F1 score across all classes as our evaluation metrics.

### 6.1 Target-based Zero-Shot Stance Detection

Target-based zero-shot stance detection (subtask A) aims to evaluate the classifier on a large number of completely unseen targets. Our experiments are performed using the full dataset with mixed targets (both noun phrases and claims), the dataset with noun-phrase targets only, and the dataset with claim targets only, respectively.

Results are shown in Table 5. First, we observe that fine-tuning MNLI pre-trained models (i.e., BART-MNLI-e$_p$ and BART-MNLI-e) consistently outperform traditional baselines (that do not use NLI pre-trained knowledge), showing the effectiveness of transforming ZSSD into the NLI task. Second, we observe that BART-MNLI-e$_p$ outperforms BART-MNLI-e, which suggests that our proposed prompts can effectively formulate noun-phrase targets into more refined hypotheses to better leverage the MNLI model for ZSSD. As we do not apply prompts to claim targets, these two models have the same performance on claim targets. Third, the BART-MNLI model without fine-tuning on EZ-Stance performs much worse than the fine-tuned BART-MNLI encoders, particularly for the noun-phrase targets. This result demonstrates the necessity of developing a large dataset for ZSSD, so that the NLI pre-trained knowledge can be better utilized. Fourth, transformer-based models outperform on claim targets in comparison to noun-phrase

| Model | | CE | WE | EdC | EnC | S | R | EP | P |
|---|---|---|---|---|---|---|---|---|---|
| BiCE | M | .441 | .443 | .480 | .451 | .458 | .485 | .465 | .439 |
| | N | .461 | .485 | .486 | .476 | .434 | .515 | .514 | .433 |
| | C | .323 | .313 | .325 | .319 | .324 | .309 | .319 | .310 |
| CrossNet | M | .482 | .489 | .501 | .484 | .470 | .531 | .489 | .484 |
| | N | .471 | .502 | .489 | .487 | .487 | .505 | .522 | .476 |
| | C | .495 | .495 | .499 | .486 | .475 | .505 | .473 | .501 |
| TGA-Net | M | .535 | .545 | .565 | .559 | .553 | .606 | .570 | .562 |
| | N | .471 | .528 | .552 | .544 | .530 | .565 | .558 | .552 |
| | C | .572 | .568 | .595 | .591 | .545 | .610 | .567 | .567 |
| BERT | M | .681 | .689 | .716 | .685 | .698 | .728 | .695 | .698 |
| | N | .567 | .560 | .580 | .577 | .587 | .612 | .578 | .569 |
| | C | .753 | .760 | .784 | .763 | .769 | .780 | .764 | .765 |
| RoBER-Ta | M | .716 | .728 | .759 | .744 | .738 | .763 | .736 | .746 |
| | N | .612 | .600 | .629 | .596 | .598 | **.633** | .625 | .591 |
| | C | .815 | .833 | .856 | .845 | .833 | **.831** | .825 | .828 |
| XLNet | M | .707 | .722 | .741 | .724 | .719 | .745 | .734 | .717 |
| | N | .586 | .609 | .596 | .588 | .581 | .622 | .605 | .580 |
| | C | .790 | .796 | .832 | .829 | .793 | .819 | .808 | .802 |
| BART-MNLI | M | .590 | .591 | .633 | .627 | .656 | .616 | .638 | .577 |
| | N | .314 | .270 | .336 | .334 | .368 | .330 | .377 | .309 |
| | C | .755 | .797 | .794 | .787 | .788 | .780 | .768 | .752 |
| **BART-MNLI-e** | M | .751 | .758 | .771 | .769 | .766 | .765 | .759 | .757 |
| | N | .604 | **.620** | **.639** | .609 | .582 | .624 | .623 | .610 |
| | C | **.850*** | **.866*** | **.874*** | **.866*** | **.866*** | .830 | **.850*** | **.846*** |
| **BART-MNLI-e$_p$** | M | **.752*** | **.769*** | **.772*** | **.771*** | **.768*** | **.783*** | **.768*** | **.763*** |
| | N | **.613** | .613 | .629 | **.613*** | **.613*** | .628 | **.638*** | **.613*** |
| | C | - | - | - | - | - | - | - | - |

Table 6: Comparison of $F1_{macro}$ of different models on EZ-STANCE subtask B. Models are trained and evaluated using datasets for 8 zero-shot domain settings. ∗: our approach improves the best traditional baseline at $p < 0.05$ with paired t-test.

targets. In contrast, BiCE and CrossNet underperform on claim targets compared to noun-phrase targets. This could be due to the transformer models' ability at capturing contextual features from claims which typically contain more contextual information than noun phrases. Last, performances on noun-phrase targets are not very high, indicating that EZ-STANCE is a very challenging new benchmark for ZSSD.

## 6.2 Domain-based Zero-Shot Stance Detection

Domain-based zero-shot stance detection (subtask B) focuses on evaluating classifiers using unseen topics from completely new domains. Particularly, we select one domain as the zero-shot domain and the rest seven domains as source domains. We train and validate models using data from source domains and test models using data from the zero-shot domain. We have eight zero-shot domain settings (each with a different zero-shot domain). As before, we experiment with the full dataset with mixed targets (M), data with noun-phrase targets (N), and data with claim targets (C), respectively.

Results are shown in Table 6. First, we notice that models show lower performance when com-

| | Subtask A | | | Subtask B (CE) | | |
|---|---|---|---|---|---|---|
| | Train | Val | Test | Train | Val | Test |
| V | 4,003 | 383 | 600 | - | - | - |
| E | 17,090 | 4,656 | 4,866 | 19,026 | 3,964 | 3,758 |

Table 7: Distribution of zero-shot targets of EZ-STANCE compared with VAST (denoted as E and V, respectively).

| Train/Val | Test | Con | Pro | Neu | All |
|---|---|---|---|---|---|
| E | V | .578 | .626 | .286 | .497 |
| V | E | .644 | .615 | .005 | .421 |
| E | E | .739 | .692 | .576 | .669 |
| V | V | .719 | .701 | .919 | .780 |

Table 8: Cross-dataset and in-dataset performance of BART-MNLI-e$_p$ trained using EZ-STANCE and VAST (denoted as E and V, respectively).

pared with the in-domain task (see results for subtask A from Table 5). This is because the domain shifts between the training and testing stages introduce additional complexity to the task, making our proposed domain-based ZSSD a more challenging ZSSD task. Second, in most cases, models perform worst on the "Covid Epidemic" domain, suggesting that the "Covid Epidemic" domain shares the least domain knowledge with other domains, making it the most difficult zero-shot domain for domain-based ZSSD. Moreover, we also observe that most models show higher performance when predicting stances for the "Rights" domain.

We also report the results for training on mixed targets and testing separately on noun-phrase targets and claim targets in Appendix H.

## 6.3 EZ-STANCE vs. VAST

We compared EZ-STANCE and VAST from two perspectives: the target diversity and the challenge of the task. In Table 7, we can observe that EZ-STANCE includes a much larger number of zero-shot targets than VAST, suggesting that models trained on EZ-STANCE can potentially be generalized to a wider variety of zero-shot targets. To understand which dataset presents more challenges, we perform the cross-dataset experiments by training our best-performing BART-MNLI-e$_p$ using one dataset and testing the model using the other dataset. We also explore the in-dataset setting by training and testing the model on the same dataset. For a fair comparison, for EZ-STANCE, we use only noun-phrase targets from subtask A.

Results are shown in Table 8. First, we observe that the model shows significantly higher performance for the in-dataset setting than the cross-dataset setting. Second, for the in-dataset

| Prompts | Con | Pro | Neu | All |
|---|---|---|---|---|
| The premise entails [target]! | .729 | .684 | **.578** | .664 |
| The premise entails the hypothesis [target]! | .727 | .688 | .571 | .662 |
| I am in favor of [target]! | .727 | .690 | .559 | .659 |
| I support [target]! | .728 | .690 | .551 | .656 |
| I am against [target]! | .718 | .678 | .558 | .652 |
| I disagree with [target]! | .721 | .686 | .567 | .658 |
| **Ours** | **.739** | **.692** | .576 | **.669** |

Table 9: Comparison of $F1_{macro}$ of BART-MNLI-e$_p$ trained using different prompts.

| Train/Val | Test | RoBERTa | BART-MNLI-e |
|---|---|---|---|
| M | N | .609 | .619 |
| M | C | .859 | .880 |
| C | N | .364 | .349 |
| N | C | .309 | .325 |

Table 10: Comparison of $F1_{macro}$ of RoBERTa and BART-MNLI-e using different targets for training/validation and test. M, N, and C are mixed targets, noun-phrase targets, and claim targets, respectively.

setting, the model trained on EZ-STANCE shows much lower performance than the one trained on VAST, particularly for the neutral class. The result demonstrates that data from the neutral class in EZ-STANCE with close semantic correlations between documents and targets are much more challenging than in VAST, where documents and targets are randomly permuted (and do not reflect the natural/real-world data for the neutral class). Third, the model trained on VAST performs extremely poorly on the neutral class of the EZ-STANCE test set, while the model trained on EZ-STANCE show much higher performance on VAST, indicating that EZ-STANCE test set captures more challenging real-world ZSSD data, especially for the neutral category. This reinforces our motivation to create a new, large dataset for ZSSD.

### 6.4 Impact of Prompt Templates

To understand the impact of prompt templates on our proposed approach, we experiment with different prompts to noun-phrase targets. Note that for prompts "*I am against [target]!*" and "*I disagree with [target]!*", we swap the stance labels between *Entailment* and *Contradiction* to ensure semantic consistency. The results are shown in Table 9, where our selected 2 prompts outperform other prompts. Also, our approach that randomly selects one prompt or the other outperforms models trained using only one type of prompt.

### 6.5 Impact of Incorporating Two Target Types

In order to explore the necessity of incorporating both noun-phrase targets and claim targets into one

| Data | RoBERTa | BART-MNLI-e |
|---|---|---|
| T+C | .853 | .879 |
| C | .526 | .541 |

Table 11: Comparison of $F1_{macro}$ of RoBERTa and BART-MNLI-e when both the tweet and claim target (T+C) are used vs. when only claim target (C) is used as the input.

dataset, we evaluate models that have been trained with noun-phrase targets using the claim targets, and vice versa. We contrast these results with models trained with the mixture of two target types and evaluated using each target type individually. Experiments are performed for subtask A, using RoBERTa and BART-MNLI-e$_p$.

From Table 10, we can observe that when models are trained using noun-phrase targets and evaluated with claim targets (and vice versa), the performance is much worse than models trained by the mixed targets. These results suggest that datasets featuring a single target type are not adept at handling the other type of target, further reinforcing the necessity of developing a dataset encompassing both target types.

### 6.6 Spuriosity Analysis for Claim Targets

We conduct a spuriosity analysis for claim targets to ensure that stance cannot be detected based solely on the claim. In Subtask A, experiments are performed on RoBERTa and BART-MNLI-e with only the claim target as input, and these results are compared with those that use both the tweet and claim target as input. From Table 11 we observe a significant performance drop when only the claim target is used as input. Therefore, the integration of tweets and claim targets is necessary for the models to accurately predict stances by learning the semantic association between them.

## 7 Conclusion

In this paper, we present EZ-STANCE, a large English ZSSD dataset. Compared with VAST, the only existing ZSSD dataset, our dataset is larger and more challenging. EZ-STANCE covers both noun-phrase targets and claim targets and also comprises two challenging ZSSD subtasks: target-based ZSSD and domain-based ZSSD. We improve the data quality of the neutral class by extracting targets from texts. We evaluate EZ-STANCE on ZSSD baselines and propose to transform ZSSD into the NLI task which outperforms traditional baselines. We hope EZ-STANCE can facilitate future research for varied stance detection tasks.

## Limitations

Our EZ-STANCE data is collected from social media. This might be perceived as a drawback as it might not encompass all facets of formal texts that could be found in essays or news comments. In the future, we aim to expand this dataset to include other text types. Yet, this restriction isn't unique to our dataset, but also affects any other datasets that concentrate on social media content.

## Ethical Statement

Our dataset does not provide any personally identifiable information. Tweets are collected using generic keywords instead of user information as queries, therefore our dataset does not have a large collection of tweets from an individual user. Thus our dataset complies with Twitter's information privacy policy.

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

# A EZ-STANCE vs. Previous English Stance Detection Datasets

We compare the statistics of our EZ-STANCE with previous English stance detection datasets in Table 12. We can observe that the sizes of existing English stance detection datasets are smaller than ours except for the WT-WT dataset (Conforti et al., 2020b). However, WT-WT is designed for in-target stance detection limited to the financial domain. In contrast, EZ-STANCE is a ZSSD dataset covering a comprehensive range of domains. When compared with rest datasets with either noun-phrase targets or claim targets, EZ-STANCE includes a much larger number of targets including both noun-phrase targets and claim targets.

# B More Examples of EZ-STANCE

In this section, we show examples of tweets with noun-phrase targets and claim targets for each domain of our EZ-STANCE dataset in Table 13.

# C Query Keywords

The full keywords set that we used for data crawling is shown in Table 14. We generate the list by gradually expanding the initial keywords set (from the Twitter *Explore* page) using the most frequent words.

| Authors | Source | # Target(s) | Target Type | Size |
|---|---|---|---|---|
| Ferreira and Vlachos (2016) | News articles | 300 | Claim | 2,595 |
| Derczynski et al. (2017) | Twitter | 305 | Claim | 5,568 |
| Gorrell et al. (2019) | Twitter, Reddit | 8,574 | Claim | 8,574 |
| Mohammad et al. (2016b) | Twitter | 6 | Noun-phrase | 4,870 |
| Swami et al. (2018) | Twitter | 1 | Noun-phrase | 3,545 |
| Conforti et al. (2020b) | Twitter | 5 | Noun-phrase | 51,284 |
| Allaway and McKeown (2020) | News Comments | 5,634 | Noun-phrase | 18,545 |
| Glandt et al. (2021) | Twitter | 4 | Noun-phrase | 6,133 |
| Li et al. (2021) | Twitter | 3 | Noun-phrase | 21,574 |
| **EZ-STANCE (ours)** | Twitter | 26,612 | Noun-phrase, Claim | 30,606 |

Table 12: Comparison of English stance detection datasets.

| | | |
|---|---|---|
| **CE** | Tweet | Cost of living off the scale, country being flooded with migrants, covid scam and jab injuries out there. How much more before the people decide enough is enough. |
| | N target/Stance | Covid Scams / Against |
| | C target/ Stance | Skyrocketing living costs and on the other side migrants will come in a lot of amounts so the country's population will increase someday. / Neutral |
| **WE** | Tweet | China's economy isn't just doing well. It is increasingly becoming 1 in several categories. Home prices are growing at slow and healthy rates, inflation is normal and healthy and the yuan is solid. The west should be trying to befriend China. Make a friend, not an adversary. |
| | N target/Stance | China's economy / Favor |
| | C target/ Stance | The economy of china is decreasing at an alarming rate due to which it's occupied last position in several categories. / Against |
| **EdC** | Tweet | To my Twitter pals who are parents in Ontario, trying to deal with homes chooling and work and all the stresses of the pandemic, my God, I don't know how you've managed to pull this off. But you have, even if you're exhausted. And you all rock. |
| | N target/Stance | home schooling / Against |
| | C target/ Stance | Parents in Ontario have managed to cope with homeschooling, work, and the pandemic, even if they are exhausted. / Favor |
| **EnC** | Tweet | Interviewer: why do you want this position? Me: so I can pay for all the online shopping I did this while being stressed about this interview. |
| | N target/Stance | online shopping / Favor |
| | C target/Stance | I do online shopping when I'm stressed. / Neutral |
| **S** | Tweet | Dwyane Wade winning an NBA Championship in his 3rd NBA season as the best player on the team .. does not get spoken on enough. |
| | N target/Stance | Dwyane Wade / Favor |
| | C target/Stance | Dwyane Wade's success in his 3rd NBA season made him the best player of all times. / Neutral |
| **R** | Tweet | The FEUHS Student Government is one with the LGBTQIA community in celebrating the PrideMonth2021 and pursuing equal rights for everyone, regardless of sexual orientation, gender identity, and expression. |
| | N target/Stance | Equal Rights / Favor |
| | C target/ Stance | Regardless of sexual orientation, gender identity, or gender expression, the FEUHS Student Government opposes equitable rights for everyone. / Against |
| **EP** | Tweet | The Sines coal plant in Portugal has been shut down nine years ahead of schedule, reducing the country s carbon emissions by 12%. A second and final plant is due to close in November which will make Portugal the fourth European country to eliminate. |
| | N target/Stance | Carbon emissions / Against |
| | C target/ Stance | Portugal's Sines coal facility was shut down nine years earlier than expected, cutting the nation's carbon emissions by 12 percent. / Favor |
| **P** | Tweet | I wish Democrats would play tough and just release an ad that says "GOP loves guns more than our kids." Just show the 234 mass shootings in 2022 and how GOP has obstructed every attempt at gun reform. There's no lie in that claim. At the very least don't call them "rational." |
| | N target/Stance | GOP / Against |
| | C target/Stance | The GOP will bring gun reform to stop the mass shootings. / Neutral |

Table 13: Examples of noun-phrase targets and claim targets for tweets in each domain of our EZ-STANCE dataset. "N target" and "C target" represent the noun-phrase target and the claim target, respectively.

YouTube shorts, modern history, work from home, herd immunity, living with covid, Fauci, public education, college football, pop culture, war, LGBTQ, environmental awareness, YouTube, career, vaccine, reels, democracy, pop culture, online shopping, hockey, reform, AI assistance writing, working class, election, parenting, global news, China, NBA, sports, student loan, traditional culture, Asian hate, presidential debate, Russia, bully, climate change, medicare, forcing electrical power, Mideast, doctors and patients, anti LGBTQ, post-covid, cooking, Snapchat, EU, presidential election, tictok, pfizer, business, general election, baseketball, prices, Chinese history, insurance, covid conspiracy, live shopping, SAT, Taliban, MLB, baseball, vaccine injury, tiger parents, environmental protection a, gency cultural output, Reels, government, family, new energy, WFH, clean energy, consumption concept, right wing, quality education, world news, stock market, private education, racism, long covid, NFL, vote, negative population growth, youtube, NASA, co-existence with Covid, WWE, DPR, political correctness, world cup, relationship, epidemic prevention, mideast, artificial intelligence, ethical consumption, Garbage classification, arming teachers, force kid to compete, health insurance, media, Negative population growth, terrorism, NATO, population aging, MLB's rule change, technology, wildfire, gun control, gender equality, migrant, doctors and patient, debate, mRNA vaccine, boxing, booster, leftists, republican, life in reels, abortion, teacher carry gun, Disney, overloaded kids, reward unreliable electricity gasoline price, international student, Ukraine, women's football, BLM, DPRK, privacy, shut down coal plants, homeschooling, physical education, men's football, NCAA, security, mask, sealed management, medical insurance, vegetarian, short video, iPhone, Iran, democrat, FDA, mid-term election, livestream shopping, CDC, women's rights, politic, electric vihicles, new york time, Hollywood, immigrant, Metoo, covid-19, equal rights, nuclear energy, mask mandate

Table 14: The full query keywords list used in our work for tweet crawling.

## D  Split Method

Initially, we randomly select x% of unique tweets for the training set and the rest as the combination of validation and test set. We then move data with overlapping targets and documents from the mixture of validation and test sets to the training set. After this step, we may introduce some additional overlapping targets during the transaction. This is because the tweets that are moved to the training set may have other noun-phrase targets that overlap with the remaining validation and test set. Therefore we repeat this transferring procedure y times until we do not have any overlapping targets and documents between the training set and the mixture of validation and test set. In our experiments, we use x=40% and y=4, because with these parameters, 66% tweets are split into our final training (similar to VAST). We then perform similar procedures to split validation and the test set. Therefore, the training, validation, and test set do not include overlapping tweets and targets with each other.

## E  Token Overlap

We also provide the average percentage of target tokens that overlap with tokens in tweets. The results are shown in Table 18. We observe that noun-phrase targets show a higher overlapping percentage than claim targets. This can be attributed to the fact that annotators tend to summarize noun-phrase targets using tokens that carry similar semantics from the text.

## F  Full Statistics of Subtask B

The statistics of the 8 dataset splits (data from seven domains for training and validation, and the data from the left-out domain as the zero-shot test set) are shown in Table 15.

## G  Training Details

Our experiments are carried out using an NVIDIA RTX A5000 GPU based on the PyTorch (Paszke et al., 2019). Hyperparameters were fine-tuned using our validation set. The BiCE and CrossNet models were trained using AdamW (Loshchilov and Hutter, 2019) as the optimizer with a learning rate of 0.001. Each model was trained for 20 epochs, with each mini-batch of size 128. As for TGA-Net, we adhered to the hyperparameters as recommended in prior research (Allaway and McKeown, 2020). The AdamW optimizer with a learning rate of 2e-5 was utilized for BERT, RoBERTa, XLNet models, BART-MNLI-e, and BART-MNLI-$e_p$ models, which were fine-tuned for 4 epochs using batch size of 64. The entire training process was completed within 3 hours. Each result is the average of 4 runs with different initializations.

## H  Evaluations on Models Trained by Mixed Targets with Noun-Phrase Targets and Claim Targets

In subtask A and subtask B, for experiments using mixed targets, we test the baseline models using noun-phrase targets and claim targets separately. Our goal is to better understand how each model trained on mixed targets performs for each type of target separately. The results for subtask A and sub-

|  |  | # Examples | | # Unique | | | Avg. Length | | |
|---|---|---|---|---|---|---|---|---|---|
|  |  | N | C | N | C | T | N | C | T |
| Covid Epidemic | Train | 8,498 | 13,167 | 5,875 | 13,151 | 4,389 | 2 | 18.6 | 39.3 |
|  | Val | 1,231 | 2,754 | 1,220 | 2,744 | 918 | 2.3 | 18.4 | 39.5 |
|  | Test | 1,716 | 2,607 | 1,156 | 2,602 | 869 | 1.9 | 18.7 | 41.1 |
| World Event | Train | 8,457 | 13,536 | 5,867 | 13,515 | 4,512 | 2 | 18.6 | 39.2 |
|  | Val | 1,211 | 2,688 | 1,197 | 2,678 | 896 | 2.4 | 18.4 | 39.7 |
|  | Test | 1,576 | 2,304 | 1,192 | 2,304 | 768 | 1.9 | 18.9 | 41.6 |
| Education and Culture | Train | 8,769 | 13,641 | 5,984 | 13,620 | 4,547 | 2 | 18.6 | 39.3 |
|  | Val | 1,248 | 2,730 | 1,234 | 2,720 | 910 | 2.3 | 18.3 | 38.8 |
|  | Test | 1,432 | 2,157 | 1,052 | 2,156 | 719 | 2 | 19.2 | 42.2 |
| Entertainment and consumption | Train | 8,535 | 13,107 | 5,750 | 13,086 | 4,369 | 2 | 18.9 | 40.3 |
|  | Val | 1,175 | 2,604 | 1,163 | 2,599 | 868 | 2.3 | 18.6 | 40.4 |
|  | Test | 1,819 | 2,817 | 1,361 | 2,812 | 939 | 1.9 | 17.3 | 35.2 |
| Sports | Train | 9,525 | 14,004 | 6,083 | 13,985 | 4,668 | 1.9 | 18.7 | 40.1 |
|  | Val | 1,213 | 2,667 | 1,203 | 2,661 | 889 | 2.4 | 18.5 | 40.2 |
|  | Test | 1,232 | 1,857 | 985 | 1,850 | 619 | 2 | 18.2 | 35.3 |
| Rights | Train | 8,541 | 13,422 | 5,809 | 13,402 | 4,474 | 2 | 18.6 | 39.4 |
|  | Val | 1,222 | 2,751 | 1,209 | 2,742 | 917 | 2.3 | 18.5 | 40.1 |
|  | Test | 1,685 | 2,355 | 1,237 | 2,350 | 785 | 2 | 18.8 | 39.7 |
| Environmental Protection | Train | 8,878 | 13,929 | 6,030 | 13,907 | 4,643 | 1.9 | 18.6 | 39.3 |
|  | Val | 1,243 | 2,757 | 1,228 | 2,747 | 919 | 2.4 | 18.3 | 39.1 |
|  | Test | 1,343 | 1,842 | 1,001 | 1,842 | 614 | 2.3 | 19.4 | 41.9 |
| Politic | Train | 8,283 | 13,236 | 5,727 | 13,217 | 4,412 | 2 | 18.6 | 39.4 |
|  | Val | 1,214 | 2,703 | 1,204 | 2,693 | 901 | 2.4 | 18.4 | 39.5 |
|  | Test | 1,837 | 2,589 | 1,342 | 2,586 | 863 | 1.9 | 18.7 | 40.3 |

Table 15: Data statistics of all 8 dataset splits for subtask B. N, C, and T represent noun-phrase targets, claim targets, and tweets, respectively.

task B are shown in Table 16 and Table 17, respectively. We can observe that most compared models show higher performance on the claim targets than the noun-phrase targets. Compared with models trained solely on noun-phrase targets and claim targets (see Table 5 and Table 6), we can observe that models trained on mixed targets show lower performance for noun-phrase targets and slightly higher performance for claim targets. This demonstrates that incorporating noun-phrase targets during training can boost the performance of claim targets. However, training with a mixture of claim targets negatively impacts the performance of noun-phrase targets. This implies a continued need for ZSSD models capable of effectively leveraging both types of targets, which we leave as our future work.

| | Mixed targets | | | | Noun-phrase targets | | | | Claim targets | | | |
|---|---|---|---|---|---|---|---|---|---|---|---|---|
| | Con | Pro | Neu | All | Con | Pro | Neu | All | Con | Pro | Neu | All |
| BiCE | .539 | .358 | .536 | .478 | .550 | .508 | .469 | .509 | .303 | .318 | .367 | .329 |
| Cross-Net | .504 | .485 | .571 | .520 | .553 | .539 | .434 | .509 | .461 | .467 | .641 | .523 |
| TGA Net | .558 | .564 | .625 | .582 | .641 | .609 | .435 | .562 | .522 | .534 | .706 | .587 |
| BERT | .724 | .732 | .756 | .738 | .682 | .634 | .493 | .603 | .744 | .786 | .878 | .803 |
| RoBERTa | .787 | .785 | .769 | .780 | .692 | .642 | .526 | .620 | .830 | .859 | .886 | .859 |
| XLNet | .767 | .766 | .760 | .764 | .679 | .652 | .469 | .600 | .808 | .834 | .883 | .842 |
| **BART-MNLI-e** | .816 | .808 | .773 | .799 | .702 | .679 | .495 | .626 | .865 | .884 | .896 | .882 |
| **BART-MNLI-e$_p$** | **.818** | **.813** | **.783** | **.805** | **.707** | **.681** | **.544** | **.644** | **.869** | **.885** | **.897** | **.884** |

Table 16: Comparison of different models in subtask A, which are trained on mixed targets and tested using the full test set with mixed targets (M), the noun-phrase targets (N), and the claim targets (C), respectively. Results are averaged over four runs.

| Model | | CE | WE | EdC | EnC | S | R | EP | P |
|---|---|---|---|---|---|---|---|---|---|
| | M | .441 | .443 | .480 | .451 | .458 | .485 | .465 | .439 |
| BiCE | N | .463 | .481 | .495 | .479 | .453 | .470 | .468 | .441 |
| | C | .324 | .319 | .329 | .331 | .294 | .312 | .340 | .298 |
| | M | .482 | .489 | .501 | .484 | .470 | .531 | .489 | .484 |
| CrossNet | N | .473 | .482 | .484 | .476 | .454 | .515 | .473 | .470 |
| | C | .491 | .485 | .502 | .483 | .467 | .513 | .485 | .486 |
| | M | .535 | .545 | .565 | .559 | .553 | .606 | .570 | .562 |
| TGA-Net | N | .495 | .522 | .558 | .539 | .547 | .559 | .566 | .534 |
| | C | .562 | .554 | .598 | .588 | .554 | .610 | .562 | .563 |
| | M | .681 | .689 | .716 | .685 | .698 | .728 | .695 | .698 |
| BERT | N | .549 | .564 | .587 | .561 | .570 | .600 | .582 | .571 |
| | C | .760 | .768 | .792 | .766 | .772 | .783 | .770 | .765 |
| | M | .716 | .728 | .759 | .744 | .738 | .763 | .736 | .746 |
| RoBERTa | N | .587 | .576 | .604 | .587 | .582 | .618 | .606 | .610 |
| | C | .798 | .824 | .855 | .847 | .825 | .830 | .816 | .824 |
| | M | .707 | .722 | .741 | .724 | .719 | .745 | .734 | .717 |
| XLNet | N | .554 | .570 | .572 | .559 | .584 | .600 | .610 | .591 |
| | C | .788 | .810 | .841 | .814 | .813 | .810 | .806 | .792 |
| | M | .751 | .758 | .771 | .769 | .766 | .765 | .759 | .757 |
| **BART-MNLI-e** | N | **.618** | .600 | .602 | **.622** | **.604** | .624 | .623 | .607 |
| | C | .837 | .861 | **.873** | .865 | .857 | .844 | .841 | .849 |
| | M | **.752** | **.769** | **.772** | **.771** | **.768** | **.783** | **.768** | **.763** |
| **BART-MNLI-e$_p$** | N | .607 | **.619** | **.615** | .619 | .599 | **.631** | **.637** | **.615** |
| | C | **.841** | **.868** | .872 | **.868** | **.863** | **.863** | **.847** | **.854** |

Table 17: Comparison of $F1_{macro}$ of different models trained on mixed targets for 8 different zero-shot domain settings, and tested using the full test set with mixed targets (M), the noun-phrase targets (N), and the claim targets (C), respectively. Results are averaged over four runs.

| | N | C |
|---|---|---|
| Train | 74.5% | 41.6% |
| Val | 77.0% | 41.3% |
| Test | 76.0% | 40.6% |

Table 18: Average percentage of token overlap between two types of targets and tweets. N and C represent noun-phrase targets and claim targets, respectively.

