# OpenReview forum: "EZ-STANCE: A Large Dataset for Zero-Shot Stance Detection"
_EMNLP/2023/Conference — EMNLP 2023 Conference Desk Rejected Submission_

### Official Review · Reviewer_6i45 · 2023-07-24

**Soundness:** 3

**Ethical Concerns:**

Yes

**Excitement:**

3: Ambivalent: It has merits (e.g., it reports state-of-the-art results, the idea is nice), but there are key weaknesses (e.g., it describes incremental work), and it can significantly benefit from another round of revision. However, I won't object to accepting it if my co-reviewers champion it.

**Justification For Ethical Concerns:**

Does the paper describe how intellectual property (copyright, etc) was respected in the data collection process?

Slightly mentioned in the ethic concerns session.

Does the paper describe how participants’ privacy rights were respected in the data collection process?

Together with the previous question, mentioned slightly in ethics concerns sesions

Does the paper describe how crowd workers or other annotators were fairly compensated and how the compensation was determined to be fair?

Not fully discussed.

Does the paper indicate that the data collection process was subjected to any necessary review by an appropriate review board?

Not mentioned.

**Missing References:**

other stance detection dataset, for example, https://aclanthology.org/2021.findings-acl.208/

**Paper Topic And Main Contributions:**

The author proposes a new dataset from Twitter in zero-shot stance detection, called EZ-STANCE. It includes noun-phrase and claim targets, with two subtasks in target and domain.

**Questions For The Authors:**

1. Related works session, why not mention any works or datasets in general stance detection, other than ZSSD. It can be seen that there are few datasets created for general stance detection, but not discussed here.

2. Some parts are not written in a very concise manner and the word choice needs to be improved. For example, where the data collection, filtering and annotation can be more concise and shorter.

3. What is the purpose to convert the ZSSD into NLI task? What is the relationship of NLI task performance and the quality of your datasets? More importantly, what is the gain of using your proposed datasets compared with the other datasets for NLI task?

**Reasons To Accept:**

1. the description, collection, filtering and annotation process is well discussed for this dataset.

2. The author also compared this dataset with mentioned only existing dataset, VAST, to demonstrate the scale and challenges.

3. The author tried to convert ZSSD into NLI task with language models and compare the performance with transformer-based models.

**Reasons To Reject:**

1. The author didn't provide a comprehensive review of the general scenarios of stance detection to zero-shot stance detection. There is no discussion of whether there are any other stance detection datasets from the general purpose, followed by ZSSD datasets.

2. The experiments of transferring ZSSD to NLI seem ok, but the comparison between transformer-based models and NLI pre-trained models that have been fine-tuned on proposed datasets is not reasonable. The author fails to explain the size and parameters of each model and why they can be compared with each other. It's not fair to conduct experiments on imbalanced models and there is no doubt that NLI models will generally do well on NLI-formatted tasks compared with other models.

3. The overall writing needs to improve, most captions of the figures are not clear and concise and the discussion of this paper is not so insightful.

4. When there is only one existing dataset available, it's hard to convince the reader by comparing it with the only existing one. Instead, the author should try to discuss with more insights and future works.

**Reproducibility:**

3: Could reproduce the results with some difficulty. The settings of parameters are underspecified or subjectively determined; the training/evaluation data are not widely available.

**Reviewer Confidence:**

3: Pretty sure, but there's a chance I missed something. Although I have a good feel for this area in general, I did not carefully check the paper's details, e.g., the math, experimental design, or novelty.

**Typos Grammar Style And Presentation Improvements:**

line 022, try to use an anonymous github link for holding your repo under review, which can give other reviews a better idea of your code and experiment settings.

Table 2, missing column title of "shot-term"

line 195, check the grammar.

start from line 230, check your "", quotation marks

Table 6, we have no idea of the meaning of MNC before we look for them in the text. Pls add explanations.

---

> ### Author Rebuttal · Authors · 2023-08-28
>
> Thank you very much for your feedback. We respond to your concerns below and we will integrate them in the paper.
>
> > Review of other stance detection works/datasets and missing reference
>
>  Indeed, in the Related Work section we focus our discussion on contrasting EZ-STANCE with VAST as both are ZSSD datasets, but we would like to point out that we did include in Table 12 in Appendix a comparison between EZ-STANCE and other previous datasets in the general stance detection (including the P-STANCE dataset that you suggested). However, we agree with you that a better flow of the paper will be to discuss the existing datasets from the general stance detection in the main paper and we will move this comparison in the related work section with more details.  Thank you for pointing this out!
>
> Regarding other general scenarios of stance detection, in the Related Work section, we start by discussing the earlier stance detection tasks, i.e., in-target stance detection and cross-target stance detection (please see lines 133-152). Then, we introduce the zero-shot stance detection (please see lines 153-175). Last, we provide the review for stance detection works using noun-phrase targets and claim targets, respectively (lines 176-192).
>
>
> > Comparison with transformer-based models
>
> Thank you for your suggestion. To better understand the effectiveness of our approach, we fine-tuned and compared the following models using noun-phrase targets of subtask A: 1) BERT and RoBERTa; 2) BERT and RoBERTa with prompts, denoted as BERT-p and RoBERTa-p, respectively; 3) BERT and RoBERTa pre-trained on MNLI (obtrained from huggingface), denoted as BERT-MNLI and RoBERTa-MNLI, respectively; 4) BERT-MNLI and RoBERTa-MNLI with prompts, denoted as BERT-MNLI-p and RoBERTa-MNLI-p, respectively; and 5) BART-MNLI-e and BART-MNLI-ep. The results are shown in the table below, where we make the following observations. First, in most cases, the models pre-trained on MNLI show better performance than the models without MNLI pre-training, which suggests that useful information from MNLI can be transferred into stance detection. Second, RoBERTa-MNLI-p and BERT-MNLI-p perform better than RoBERTa-MNLI and BERT-MNLI, respectively. This result is consistent with the result for BART-MNLI-e/ep and demonstrates the effectiveness of our prompts on MNLI pre-trained models for RoBERTa and BERT. Third, BERT-p and RoBERTa-p show slight improvement over their non-prompt counterparts. This implies that our prompt is also effective for transformer models without MNLI pre-training. Last, our proposed approach achieves the best performance on BART-MNLI-ep.
>
> The number of parameters for BERT-based models (including BERT, BERT-p, BERT-MNLI, BERT-MNLI-p), RoBERTa-based models  (including RoBERTa, RoBERTa-p, RoBERTa-MNLI, RoBERTa-MNLI-p), XLNet, and BART-MNLI-e-based models (including BART-MNLI-e and BART-MNLI-ep) is 110M, 125M, 110M, and 203M, respectively. We will add more explanations on the model size in our final version. Thank you for your suggestion!
>
>
>
> |     Models | Con | Pro | Neu | All |
> |---|---|---|---|---|
> | BERT | 0.669 | 0.619 | 0.535 | 0.608 |
> | BERT-p | 0.682 | 0.616 | 0.532 | 0.610 |
> | BERT-MNLI | 0.675 | 0.642 | 0.510 | 0.609 |
> | BERT-MNLI-p | 0.681 | 0.643 | 0.530 | 0.618 |
> | RoBERTa | 0.712 | 0.677 | 0.529 | 0.639 |
> | RoBERTa-p | 0.714 | 0.678 | 0.528 | 0.640 |
> | RoBERTa-MNLI | 0.714 | 0.677 | 0.532 | 0.641 |
> | RoBERTa-MNLI-p | 0.717 | 0.684 | 0.535 | 0.645 |
> | BART-MNLI-e | 0.729 | 0.690 | 0.542 | 0.653 |
> | BART-MNLI-ep | 0.739 | 0.692 | 0.576 | 0.669    |
>
> Table 1: Comparison of different models trained using noun-phrase targets for EZ-STANCE subtask A. The performance is reported using F1 score for against (Con), favor (Pro), neutral (Neu), and the F1-macro (All).
>
> > Paper writing
>
> Thank you for pointing this out. We will improve the clarity throughout the paper and will make the writing more concise. We will also improve figure captions and typos in the final version.
>
>
> > Comparison with only VAST and adding more future directions
>
> Thank you for your suggestion. We will add more discussion on what future directions our dataset can offer and its full capability. We also want to highlight that EZ-STANCE is not a simple newer version of the VAST dataset. Instead, EZ-STANCE is a comprehensive ZSSD dataset that includes the more challenging task (domain-based ZSSD) and more diverse target types (claim targets) that are not covered by the VAST dataset. Our experimental results support the necessity of including the additional task and the target type (please see Sections 6.2 and 6.5, respectively). Moreover, compared with VAST, our method of data annotation can improve data quality (particularly for the neutral class) and make stance detection more challenging (Section 6.3).
>
>
> > What is the purpose of converting the ZSSD into the NLI task? What is the relationship of NLI task performance and the quality of your datasets? More importantly, what is the gain of using your proposed datasets compared with the other datasets for the NLI task?
>
> The reason we transform ZSSD into NLI is that the two tasks have some similarities between them–an aspect which was majorly overlooked before and represents a novel component of our work. Hence, by converting ZSSD into NLI, we can make use of the large amounts of data annotated for NLI (e.g., MNLI contains ~400,000 samples). Our intuition was that we can transfer knowledge from NLI, and thus, can improve the performance of ZSSD. Indeed, the results of our experiments validate our intuition.
>
> To verify the benefit of using only MNLI (as we proposed) compared with other datasets for NLI, we ran an experiment on noun-phrase targets where we compared BART-MNLI-ep with a BART model pre-trained on the combination of several NLI datasets, i.e., SNLI, MNLI, FEVER-NLI, and ANLI that is available from huggingface and that we further fine-tuned on our dataset. We call this model BART-COMBINATION-ep. The F1-macro of BART-MNLI-ep vs. BART-COMBINATION-ep is 0.669 vs. 0.670. We can see that they perform similarly, which means the amount of information transferred from MNLI alone or from the combination of several NLI datasets is similar.
>
>
> > Anonymous GitHub link
>
> Thank you for your suggestion. Please check the following  GitHub link for our code and experiment settings: https://github.com/yeye8863/EMNLP23_code.
>
>
>
> > Ethical Concerns:
>
> Our work is covered by an IRB exemption.

---

### Official Review · Reviewer_zFVb · 2023-08-04

**Typos Grammar Style And Presentation Improvements:** The English expression needs to be im…
**Soundness:** 3

**Excitement:**

4: Strong: This paper deepens the understanding of some phenomenon or lowers the barriers to an existing research direction.

**Missing References:**

no

**Paper Topic And Main Contributions:**

This paper presents a large English ZSSD dataset with 30,606 annotated text-target pairs EZ-STANCE. EZ-STANCE includes both noun-phrase targets and claim targets, covering a wide range of domains. The authors provide an in-depth description and analysis of our dataset. We evaluate EZ-STANCE using state-of-the-art deep learning models. The authors further propose to transform ZSSD into the NLI task by applying two simple and effective prompts to noun-phrase targets. Experimental results show that EZ-STANCE is a challenging new benchmark, which provides significant research opportunities on ZSSD.

**Questions For The Authors:**

1. Are some pseudo-labeling methods used when building data sets?
2. What's special about the dataset compared with other existing ZSSD data sets?


**Reasons To Accept:**

1. This paper presents a large English ZSSD dataset with 30,606 annotated text-target pairs EZ-STANCE. EZ-STANCE is a challenging new benchmark, which provides significant research opportunities on ZSSD.
2. The authors provide an in-depth description and analysis of our dataset and introduce two challenging subtasks for ZSSD: target-based ZSSD and domain-based ZSSD.


**Reasons To Reject:**

1. Are some pseudo-labeling methods used when building data sets?
2. What's special about the dataset compared with other existing ZSSD data sets?


**Reproducibility:**

4: Could mostly reproduce the results, but there may be some variation because of sample variance or minor variations in their interpretation of the protocol or method.

**Reviewer Confidence:**

4: Quite sure. I tried to check the important points carefully. It's unlikely, though conceivable, that I missed something that should affect my ratings.

---

> ### Author Rebuttal · Authors · 2023-08-28
>
> Thank you very much for your feedback and affirmation of our paper.
>
> > Pseudo-labeling method used when building data sets?
>
> No, we did not use any pseudo-labeling methods when building the dataset.
>
>
>
> > EZ-STANCE compared with other existing ZSSD data sets
>
> We compare below EZ-STANCE with VAST which is the only other existing English ZSSD dataset: (1) VAST contains only noun-phrase targets whereas EZ-STANCE covers both noun-phrase targets and claim targets; (2) VAST is developed for the target-based ZSSD task. In EZ-STANCE, we not only include the target-based ZSSD, but also propose the more challenging domain-based ZSSD task; (3) VAST generates data for the neutral class by randomly permuting existing texts and targets, leading to a lack of semantic correlation between the two. Comparatively, for EZ-STANCE, annotators manually extract targets from each tweet, ensuring semantic relevance to the tweet content; and (4) Lastly, EZ-STANCE is 1.9 times larger than VAST.
>
> In addition, please note that we included in Table 12 in the Appendix a comparison of EZ-STANCE with previous English stance detection datasets.

---

### Official Review · Reviewer_rLV2 · 2023-08-04

**Soundness:** 4

**Excitement:**

4: Strong: This paper deepens the understanding of some phenomenon or lowers the barriers to an existing research direction.

**Paper Topic And Main Contributions:**

The authors construct a new dataset for zero-shot stance detection. The dataset has two types of targets, noun phrases and claims, unlike existing ZSSD data which only has noun phrases. There is also a subtask where test targets are not only new but also from a new domain. To construct instances, the authors scrape tweets using keywords that they split into 8 domains. They then have human annotators provide the targets and corresponding stance labels. Human annotators have good agreement. The authors also provide benchmark results on their dataset and conduct interesting analysis into the results.

Main contributions: new dataset

**Questions For The Authors:**

- A: 3.3.1: How do you validate the assignment of the noun-phrase targets from step 1? In step 2, what happens if the assigned topic is not relevant or there is not enough information to determine stance? Are these all just assigned neutral?
- B: 3.3.2: The construction of the favor claims seems like it would just lead to paraphrases of the tweet. How often does this happen?
- C: It seems like the inclusion of "entails" in the hypothesis might bias the model towards predict entailment. Did you notice this at all?
- D: L455 - did you try fine-tuning BERT or RoBERTa pretrained on MNLI? Did you see the same kind of improvements there as with BART?
- E: It is surprising to me that the prompts do not have more of an impact of the model performance (BART-MNLI-e vs. BART-MNLI-ep). Do you have any hypotheses for why this might be happening? The small change in results suggests the model may be picking up on some clues based on the similarity of the target to the tweet. Did you check how often the target actually appears in the tweet and how the performance might differ on those instances versus others?



**Reasons To Accept:**

- The dataset will be important for future improvements on zero-shot stance detection.
- The analysis is interesting and clearly shows the difficulty of the dataset and the improvements in it over prior datasets.

**Reasons To Reject:**

None

**Reproducibility:**

4: Could mostly reproduce the results, but there may be some variation because of sample variance or minor variations in their interpretation of the protocol or method.

**Reviewer Confidence:**

5: Positive that my evaluation is correct. I read the paper very carefully and I am very familiar with related work.

**Typos Grammar Style And Presentation Improvements:**

- L223: Can you include a list of the single stance topics in the appendix?
- Table 6: please put the definitions of the M, N, and C abbreviations in the table caption to make it easier to read. Also, please include in the caption a reference to somewhere listing the domain abbreviations. If there is space in the final version, I would consider making table 6 a two-column table (potentially flip the rows an columns) so the domain names can actually be spelled out.
- Table 8 is a little hard to read. Consider putting a line separating the top two results (the cross-dataset) from the bottom two (the in-dataset) so it is easier to understand.

---

> ### Author Rebuttal · Authors · 2023-08-28
>
> Thank you for your positive and constructive feedback and review!
>
> >Noun-phrase targets validation - assigning to neutral
>
> In step 1 we instructed our annotators to identify stance-rich noun-phrase targets and to avoid targets that are stance-irrelevant (to the extent possible).  However, in case irrelevant targets are identified in step 1, annotators in step 2 are indeed instructed to annotate them as neutral. We will make this clear in the final version and thank you for pointing this out!
>
> >Favor claims being paraphrases?
>
> We randomly selected 100 favor claims from the test set and asked a graduate student with expertise in stance detection to perform a human evaluation. We found that  49 of the 100 selected favor claims are paraphrases of the tweets. However, we observed that the paraphrased claims possess many non-overlapping vocabulary words and have different sentence structures, rather than being just simple copies of the tweets with minor word replacements.
>
>
> >Adding “Entails” in the hypothesis might bias the model to predict entailment?
>
> We looked into potential biases caused by various prompts vs. no prompt at all (i.e., using just the target itself) and we observed slight biases. We show these results in Table 1 below. As we can see, when we use “entails” in the prompt, the model (BART-MNLI-ep) has a slight bias towards the entailment class.
>
> | Prompts | Contradiction | Entailment | Neutral |
> |---|---|---|---|
> | Prompt with “entails” | 0.30 | 0.38 | 0.32 |
> | Prompt with “contradicts” | 0.37 | 0.30 | 0.33 |
> | No prompt | 0.32 | 0.35 | 0.33 |
>
> Table 1: Fraction of examples from the test set predicted in each class when prompts include “entails”, “contradicts” and no prompt (i.e., just the target), respectively.
>
> >Did you try fine-tuning BERT or RoBERTa pre-trained on MNLI? Did you see the same kind of improvements there as with BART?
>
> Thank you for your suggestion. To better understand the effectiveness of our approach on other MNLI pre-trained models, we fine-tuned and compared the following models using noun-phrase targets of subtask A: 1) BERT and RoBERTa; 2) BERT and RoBERTa with prompts, denoted as BERT-p and RoBERTa-p, respectively; 3) BERT and RoBERTa pre-trained on MNLI (obtrained from huggingface), denoted as BERT-MNLI and RoBERTa-MNLI, respectively; 4) BERT-MNLI and RoBERTa-MNLI with prompts, denoted as BERT-MNLI-p and RoBERTa-MNLI-p, respectively; and 5) BART-MNLI-e and BART-MNLI-ep. The results are shown in the table below, where we make the following observations. First, in most cases, the models pre-trained on MNLI show better performance than the models without MNLI pre-training, which suggests that useful information from MNLI can be transferred into stance detection. Second, RoBERTa-MNLI-p and BERT-MNLI-p perform better than RoBERTa-MNLI and BERT-MNLI, respectively. This result is consistent with the result for BART-MNLI-e/ep and demonstrates the effectiveness of our prompts on MNLI pre-trained models for RoBERTa and BERT. Third, BERT-p and RoBERTa-p show slight improvement over their non-prompt counterparts. This implies that our prompt is also effective for transformer models without MNLI pre-training. Last, our proposed approach achieves the best performance on BART-MNLI-ep.
>
> |     Models | Con | Pro | Neu | All |
> |---|---|---|---|---|
> | BERT | 0.669 | 0.619 | 0.535 | 0.608 |
> | BERT-p | 0.682 | 0.616 | 0.532 | 0.610 |
> | BERT-MNLI | 0.675 | 0.642 | 0.510 | 0.609 |
> | BERT-MNLI-p | 0.681 | 0.643 | 0.530 | 0.618 |
> | RoBERTa | 0.712 | 0.677 | 0.529 | 0.639 |
> | RoBERTa-p | 0.714 | 0.678 | 0.528 | 0.640 |
> | RoBERTa-MNLI | 0.714 | 0.677 | 0.532 | 0.641 |
> | RoBERTa-MNLI-p | 0.717 | 0.684 | 0.535 | 0.645 |
> | BART-MNLI-e | 0.729 | 0.690 | 0.542 | 0.653 |
> | BART-MNLI-ep | 0.739 | 0.692 | 0.576 | 0.669    |
>
> Table 2: Comparison of different models trained using noun-phrase targets for EZ-STANCE subtask A. The performance is reported using F1 score for against (Con), favor (Pro), neutral (Neu), and the F1-macro (All).
>
>
> > Prompt effects: BART-MNLI-e vs. BART-MNLI-ep
>
> Thank you for pointing this out. We would like to highlight that BART-MNLI-ep (i.e., BART-MNLI with prompt) yields a significant increase of 1.6% in F1-macro over BART-MNLI-e (i.e., BART-MNLI without prompt) on noun phrase targets (please see Table 5). That is, BART-MNLI-e achieves an F1-macro of 0.653, whereas BART-MNLI-ep achieves an F1-macro of 0.669. For mixed targets, the improvement is smaller (0.6% on F1-macro). This is because we only apply prompts to noun-phrase targets, and the enhancement brought by the noun-phrase targets is diluted by the claim targets. We will clarify this in the paper.
>
>
> > List of single stance topics
>
> We will include the complete list of single-stance topics in the appendix in our final version. Some examples are: crime and delicious food toward which most people hold against and favor stances, respectively.
>
> > Table caption abbreviations
>
> We will add the definitions of abbreviations in Table 6 and improve the table presentation per your suggestion in the final version. We will also add the separation line for Table 8 in our final version. Thank you for your suggestions!

---

### Meta-Review · Area_Chair_zsfW · 2023-09-20

**Recommendation:** 4

**Metareview:**

The paper introduces a new dataset for zero-shot stance detection called EZ-STANCE which includes both noun-phrase and claim targets, and it is more comprehensive than previous datasets.
A good set of experiments and analyses is performed on this dataset.
The reviewers all found it a helpful dataset for different research opportunities.
The authors reported new results in the rebuttal which should be included in the next version if accepted.

---

### Decision · Program_Chairs · 2023-10-07

**Decision:**

Accept-Findings

**Comment:**

The paper introduces a new dataset for zero-shot stance detection called EZ-STANCE which includes both noun-phrase and claim targets, and it is more comprehensive than previous datasets.
A good set of experiments and analyses is performed on this dataset.
The reviewers all found it a helpful dataset for different research opportunities.
The authors reported new results in the rebuttal which should be included in the next version if accepted.